# M-IDAS: Multi-modal Intrusion Detection and Analytic System

## Abstract

The analysis of modern intrusion often needs to consider the correlation between patterns from different channels such as network traffic, host behavior, and device status to achieve accurate intrusion detection. However, existing research predominantly employs single-modal data for intrusion detection & explanation, a method that, though operationally straightforward, provides constrained representational capacity for complex cases. How to leverage multi-modal fusion representations for intrusion detection and interpretation from diverse data channels remains a key challenge. In this paper, we propose a new cross-domain multi-modal intrusion detection model called **M**ulti-modal **I**ntrusion **D**etection and **A**nalytic **S**ystem (**M-IDAS**), which is based on bidirectional encoder representations from transformers. This model employs modal fusion to unify different intrusion data and pre-trains attack behavior context representations from extensive unlabeled multi-modal fused data. The pre-trained model can be fine-tuned with minimal labeled data specific to tasks, achieving state-of-the-art performance across various intrusion detection scenarios. Notably, through an analysis of model attentions during detection, we provide traceability and interpretative insights into network attack behaviors, offering a profound understanding of the network attack process.

## 1 Introduction

In an era characterized by the rapid proliferation and ubiquity of Internet of Things (IoT) devices (Jin et al., 2022; Ozmen et al., 2022; Wang et al., 2023), from smart homes to industrial control systems, ensuring intrusion detection on IoT system has never been more crucial. Recently, traffic encryption technology has been widely used to protect users' privacy and data security (Fu et al., 2023). It also brings great challenges to traffic attack detection since the intrusion traffic and attackers can evade the Network Intrusion Detection System (NIDS) (Liao et al., 2013) by stealth-enhancing and privacy-enhanced encryption technologies (Lee et al., 2020), such as TLS (Dierks & Allen, 1999; Dierks & Rescorla, 2008), VPN (Praveen et al., 2021), etc. Besides, cybercriminals also try to evade the Host Intrusion Detection System (HIDS) (Ribeiro et al., 2020a;b) by process hiding technologies, such as hooking system calls and forging process names. Those techniques potentially provide a veil for attackers and malicious software to obscure their communication and operational activities (Eskandari et al., 2020). Consequently, several researchers have explored multi-modal intrusion detection approaches (Spinoulas et al., 2021; Gomez-Barrero et al., 2018), including those based on multi-modal fingerprints, multi-modal sensors data (Wagan et al., 2023), These multi-modal techniques encompass both general attack scenarios (Xu et al., 2023) and industrial cyber-attacks (Bahadoripour et al., 2023).

Through extensive study, we observed attackers frequently disguise malicious traffic as regular HTTP/TLS traffic using encrypted network flows (Yeh et al., 2020). Single network data detection methods are incapable of effectively detecting these encrypted network attacks, as depicted in Figure 1 (a). Besides, the capability of the single host data detection method is limited, as depicted in Figure 1 (b). For example, Individual devices participate in only a small portion of attacks in DDoS attacks, which are very covert and difficult to detect.

Traditional intrusion detection methods, reliant on a singular data source, encounter significant limitations. They struggle to cross-verify malicious activities, leading to an incomplete understanding of the attack landscape and increased vulnerability to evasion by sophisticated threats (Moustafa

Figure 1: A brief illustration of the multi-modal intrusion detection and analytic system

et al., 2018; Dong et al., 2019; Ibitoye et al., 2019). This limitation stems from the inadequacy of individual data sets to comprehensively depict cyber-attack trends (**CH1**). Recent interest in incorporating multi-source data, including network traffic, system logs, and device status from distinct sources, has grown (Nandy et al., 2022; Alsaedi et al., 2020). However, issues of modality entanglement and conflict within multi-modal models (Li et al., 2022a) have not received sufficient attention. Integrating multiple modalities introduces complexities, leading to overlapping or conflicting information across different data types, presenting a challenge of modality conflict and entanglement in data under the same attack (**CH2**). While multi-modal learning has shown breakthroughs in various fields (Zhang et al., 2020; Feng et al., 2020), its application in intrusion detection remains limited. Existing work, although applying multi-modal features, often focuses solely on network traffic data, lacking the breadth required for comprehensive attack description and cross-modal interpretation capabilities (**CH3**).

Addressing the challenges outlined, we introduce M-IDAS, a novel multi-modal intrusion detection and analytic system. The system is designed to learn fused multi-modal feature representations and conduct cross-domain interpretative analytics, as depicted in Figure 1 (c). To enhance data representation robustness and resolve **CH1**, we executed fifteen categories of IoT attacks in a controlled setting, collecting data across four modalities spanning six domains. For handling multi-modal entanglement and addressing **CH2**, we propose a multi-channel convolutional encoder network approach, ensuring consistent feature representation. Finally, to tackle **CH3**, we present a bidirectional encoder classification model with a multi-head attention mechanism for intrusion detection. This model concurrently utilizes an attention score for cross-domain interpretative analysis. The effectiveness is validated through 15 attack detection tasks across six categories, achieving an average F1 of 97.7%. Ground truth construction for each attack type verifies cross-domain interpretability, with an average accuracy of 94.4%. The main contributions of this paper are summarized as follows:

**(1)** We summarize a comprehensive taxonomy for IoT-related intrusion and implement an attack replay toolkit[1], collected 15 categories of attacks across six domains at 4.52 GB.
**(2)** We propose a pre-training approach (M-IDAS) for intrusion detection. It utilizes multiview features, leveraging multi-modal fusion to learn attack feature representation across multiple domains. Employing large-scale unlabeled intrusion data, M-IDAS learns the relationship among the fused features achieving an impressive average accuracy at 98.3% across diverse tasks.
**(3)** We newly propose cross-domain multi-modal intrusion traceability and explainability analytics, which extracts key fusion features and dependency trajectories from the attention score and offers insights into which domain influences the decision-making process.

## 2 ATTACK TAXONOMY AND MULTIVIEW DATA COLLECTION

### 2.1 ATTACK TAXOMOMY

By analyzing IoT-related attack vectors, we prioritized six primary categories that cumulatively represent approximately 90% of all documented incidents (IBM-Security, 2022). Notably, DDoS attacks, particularly those leveraging the Mirai botnet infrastructure, emerged as the most dominant, constituting an estimated 40%-50% of the total IoT attack landscape. Based on this observation, our

---

[1]The toolkit and data will be released on our GitHub repository https://github.com/m-idas/M-IDAS to encourage further intrusion detection research among the AI4Security community after peer review.

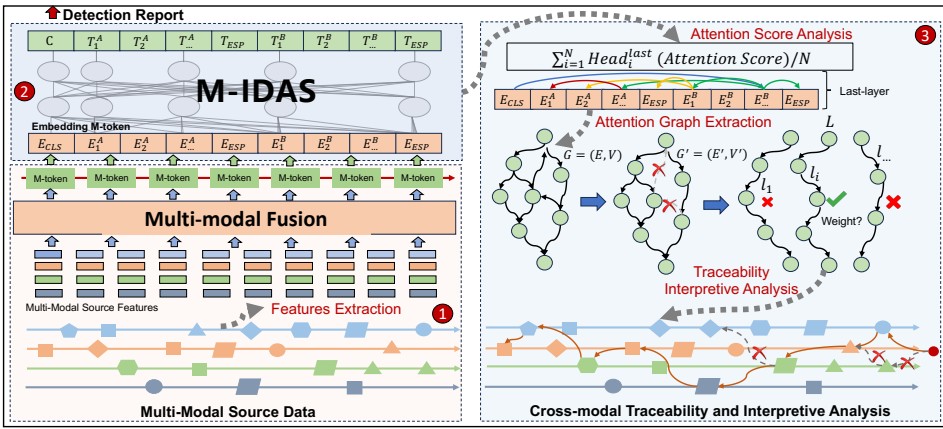

Figure 2: The overview of M-IDAS

subsequent experimental methodology involved the selection of 10 distinct DDoS attack instances coupled with 5 additional network attack vectors as representative samples for further study[2].

## 2.2 MULTIVIEW DATA SELECTION

In light of the observations across six categories of attacks, we collect data from four modalities spanning six domains: network traffic, host system event logs, host system performance metrics, and IoT device status, for intrusion detection[2]. For instance, Mirai is a prominent malware variant specifically designed to compromise and commandeer IoT devices, transforming them into part of a botnet. Upon infection, the compromised device initiates the Mirai process and subsequently targets other IoT devices once it receives specific directives from the primary attacker. The attack mechanisms predominantly utilize network protocols, with DDoS attacks being a typical manifestation. Observations from such attack instances reveal discernible patterns across various data modalities, including active processes, file operation logs, system health metrics (such as CPU and memory usage), network traffic statistics, and the operational statuses of other IoT devices.

## 3 APPROACH

In this section, we delineate the methodology encompassing feature extraction and fusion, pre-training of the intrusion detection model, fine-tuning for specific downstream tasks, and the subsequent interpretative analysis.

## 3.1 OVERVIEW OF M-IDAS

In this paper, our objective is to derive a comprehensive multi-modal intrusion data representation and subsequently detect them across varied contexts, such as DDoS attacks, IP/Port scans, and man-in-the-middle attacks, among others. Our proposed pre-training methodology encompasses three pivotal phases as shown in Figure 2. (1) *Multi-modal Feature Extraction and Fusion:* This initial phase involves the extraction and fusion of features from multi-modal intrusion data obtained from various sources such as network traffic, system logs, and IoT device statuses. The result is a set of fused features representing the multi-modal intrusion data. (2) The model training phase contains two stages, i.e., pre-traing and fine-tuning. *Pre-training for Generic Multi-modal Intrusion Data Representation:* Utilizing large-scale unlabeled multi-modal intrusion data, this phase produces generic contextual representations of multi-channel intrusion data. *Fine-tuning for Specific Downstream Tasks:* With the aid of target-specific labeled data (e.g., network traffic, process information, and IoT device status), the model undergoes fine-tuning to predict the respective categories. (3) *Traceability and Interpretive Analysis via Attention Dependency:* In this final stage, leveraging

---

[2]Details in Appendix A.1

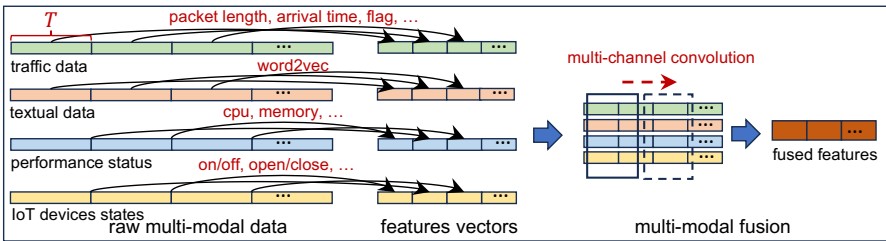

Figure 3: Multi-modal features fusion

the attention scores, the analytical model delineates the intrusion attack trajectory and pinpoints the salient features instrumental for detection.

The core architecture of M-IDAS is anchored on multiple layers of bi-directional Transformer blocks. Each of these blocks integrates multi-head self-attention layers, adept at discerning the latent correlations within the fused multi-modal intrusion data. In the present study, the architectural framework encompasses 12 Transformer blocks, each equipped with 12 attention heads within their respective self-attention layers. The dimensionality of each input token, denoted as $H$, is fixed at 768, while the total count of input tokens stands at 512.

## 3.2 M-TOKEN: MULTI-MODAL DATA FUSION

### 3.2.1 FEATURES EXTRACTION

After gathering multi-modal data, we accumulate information from various sources, including network traffic, system event logs, performance metrics, and the status of IoT devices. These data sources yield distinct feature sets: 80 dimensions from network traffic, 35 dimensions from system event logs, 23 dimensions from performance metrics, and 15 dimensions from IoT device status[3].

**Network Traffic:** This category encompasses the extraction of statistical features from network traffic data, such as duration, packet count, and packet length. It also includes the analysis of packet flag features like SYN and RST flags.
**Host System Event Logs:** Within this category, we examine operational data from host systems, including logs of running processes, executed commands. Features are derived from process names, command names. Word2Vec is used to convert process names into representative vectors, which are then combined to create comprehensive process features.
**Host System Performance Metrics:** To gauge the status of the host system, we utilize metrics related to CPU and memory usage, including absolute usage and relative percentage consumption. These metrics are condensed into a feature vector to offer a holistic view of the performance.
**IoT Device Status:** In this category, we focus on monitoring the various states of IoT devices, including connectivity status, battery capacity, binary states (e.g., window sensors), and quantitative readings (e.g., temperature sensors). A feature vector is employed to represent these states, with each column corresponding to a specific device attribute.

### 3.2.2 FEATURE FUSION

Initially, we synchronize the four categories feature timestamps across the six data channels to ensure consistent temporal alignment. This synchronization facilitates simultaneous learning of data characteristics across different modalities, addressing the limitation of intrusion detection capabilities inherent to a single modality. Employing time $T$ as the fundamental unit of data representation features corresponding to network traffic, process behavior, execution command patterns, and file operations are extracted within each interval $T$, where $T$ is set to 10 ms. Concurrently, the system's status and IoT device states are also characterized at every interval $T$, enabling the acquisition of feature representations across varying modal data. For each modality, features are encapsulated in vectors with a dimensionality of 128. As shown in Figure 3, upon extracting features across the distinct modalities, a feature vector of dimensions $6 \times 128$ is derived. To facilitate simultaneous

---

[3]The features details and description can be found in the appendix[2] and on our website[4].

feature fusion, data from each modality is first normalized. This normalized data is then processed through a convolutional autoencoder, aiding in the integration of the modalities. The resultant intermediate vector, generated post-fusion, serves as the input to the subsequent analytical model. This methodology ensures a uniform representation of disparate modal data, bolstering the detection feature's capacity to characterize potential attacks. It's noteworthy that the input dimension for the convolutional network encoder is $6 \times 128$, while the output is dimensioned at $1 \times 4$.

### 3.2.3 STRUCTURAL TOKENIZATION

We collect multi-modal data samples at intervals of $T$. After feature extraction and fusion, a fused multi-modal feature sequence is obtained, with each fused F-Vector being $1 \times 4$. We represent the F-Vector as a $token$ consisting of 32 hexadecimal characters. For training tasks, special tokens [CLS], [SEP], [PAD], and [MASK] are introduced. The initial token is always [CLS], and the final hidden layer state associated with it represents the entire sequence for classification tasks. [PAD] is used for padding to meet the minimum length requirement. The M-token sequence detected is separated by [SEP], and [MASK] appears during pre-training to grasp the context of multi-modal features. In Figure 2, we equally divided a window into two sub-windows for the Next M-token Sequence Prediction (NMSP) task. We differentiate the sub-windows by the special token [SEP] and the segment embedding indicating whether it belongs to segment A or segment B. We denote segment A as sub-window A and segment B as sub-window B respectively.

**Token Embedding:** The content of the M-token is encoded, yielding $E_{M-token}$, and $E_{M-token} \in R^H$, where the dimension $H$ is set to 768.
**Position Embedding:** Recognizing the relevance of attack behaviors to sequence order, we incorporate positional embeddings to enable the model to capture the temporal relationships of M-tokens through their relative positions. An $H$-dimensional vector $E_{pos} \in R^H$ is assigned to each input M-token to represent its position information in the sequences, where $H$ set to 768.
**Segment Embedding:** Within a window, we evenly divide the sequence into two segments, and the segment embedding of the sub-window is denoted as $E_{seg} \in R^H$, where $H$ is set to 768.

### 3.3 PRE-TRAINING M-IDAS

### 3.3.1 MMM: MASKED M-TOKEN MODEL

The Masked M-token Model (MMM) task bears resemblance to the Masked Language Model (MLM) employed by BERT. A notable distinction is the integration of attack M-tokens, which lack explicit semantics, into M-IDAS to discern dependencies among fused multi-modal features. In the pre-training phase, each M-token within the input sequence is subjected to a 15% probability of random masking. For a selected M-token, there is an 80% likelihood of substitution with [MASK], while there's a 10% probability for either its replacement with a random token or retaining it unchanged. For M-tokens that are masked, they are substituted with the designated token [MASK]. M-IDAS is then trained to infer tokens at these masked positions utilizing the surrounding context. Leveraging the profound bi-directional representation afforded by this procedure, we introduce random masking to $k$ tokens within the input sequence $X$. The loss function employed is the negative log-likelihood, which we formally define as follows:

$$Loss_{MMM} = -\sum_{i=1}^{k} \log \left( P \left( MASK_i = token_i \mid \bar{X}; \theta \right) \right) \quad (1)$$

where $\theta$ denotes the collection of trainable parameters inherent to M-IDAS. The probability $P$ is characterized by the encoder parameterized by $\theta$. $\bar{X}$ signifies the representation of $X$ post-masking, and $MASK_i$ designates the masked token situated at the $i^{th}$ position within the token sequence.

### 3.3.2 NMSP: NEXT M-TOKEN SEQUENCE PREDICTION

Our objective is to enhance the learning of multi-modal feature representations by discerning the correlations among M-tokens. The dependencies between M-tokens within the detection window are ascertained through the Next M-token Sequence Prediction (NMSP) task. In this context, a binary classifier is employed to determine if two sub-sequences originate from the same window.

Specifically, for each pair comprising sub-sequences A and B, there's a 50% probability that sub-sequence B is the immediate successor of sub-sequence A and a 50% likelihood that it is a randomly selected sub-sequence from a different window. Given an input that includes the sub-sequence pair $W_j = (sub - W_j^A, sub - W_j^B)$ and its corresponding ground-truth label $y_j$ from the set $[0, 1]$, where 0 denotes paired sub-sequences and 1 indicates unpaired ones.

$$Loss_{NMSP} = -\sum_{j=1}^{n} \log \left( P\left( y_j \mid W_j; \theta \right) \right) \tag{2}$$

The ultimate pre-training objective consolidates the two aforementioned losses:

$$Loss = Loss_{MMM} + Loss_{NMSP} \tag{3}$$

### 3.4 FINE-TUNING M-IDAS

Fine-tuning is advantageous for downstream detection tasks for several reasons: (1) The pre-trained representation is agnostic to specific intrusion classes, rendering it applicable to any class of intrusion representation. (2) The distinct [CLS] token, emanating from the output of the pre-trained model, encapsulates the representation of the entire input, amalgamating multi-modal features, and is directly amenable for classification purposes. (3) The attention parameters offer a mechanism to elucidate attack interpretability. (4) In the process of model adaptation, fine-tuning necessitates a markedly reduced quantity of labeled data relative to the voluminous dataset essential for pre-training. Concurrently, the temporal commitment for fine-tuning is notably brief compared to the protracted period demanded by pre-training. The computational expense of fine-tuning is notably lower than that of pre-training, and a singular GPU is adequately equipped to handle the task.

### 3.5 INTERPRETIVE ANALYTICS

#### 3.5.1 ATTENTION-DEPENDENCE GRAPH CONSTRUCTION

For a given sample undergoing detection, it is feasible not only to ascertain its classification outcome but also to obtain the attention scores across each layer. These scores epitomize the attention or interdependence of each token in relation to other contextual tokens, providing invaluable insights for the traceability and interpretative analysis of attacks. We derive the attention scores from each head in the terminal layer and compute their mean to represent the attention score of individual tokens. This procedure facilitates the generation of an attention-dependence weighted relationship graph $G(E, V)$, as depicted in Figure 2. In this representation, each node corresponds to a token, while the edges signify the respective attention scores.

#### 3.5.2 DEPENDENCY PATHS COMPUTATION

Given the chronological nature of network attacks, our objective is to perform source tracing and explanatory analysis by examining the dependency graph consisting of tokens. Therefore, we start by eliminating the directional edges in the dependency graph that originate from a token node at an earlier time and point to a token node at a later time. Thus, we obtain a time-unidirectional dependency graph denoted as $G^{'}(E^{'}, V^{'})$. Following the acquisition of the relationship graph, we employ a depth-first algorithm to navigate the [CLS] pair graph and extract the set of all dependency paths denoted as $L$. Subsequently, we employ a weighted method to compute the cumulative attention scores for each dependency path $l$ in the set (see Equation 4), ultimately selecting the path $\delta$ with the highest sum of attention scores as the primary attack pathway (see Equation 5).

$$AttentionScore_l = \sum_{i=1}^{M-1} v_{i,i+1}, \ where \ v_{i,i+1} \in l, \ |\ l\ | = M. \tag{4}$$

$$AttackPath = \ \delta \mid AttentionScore_\delta = \max\{AttentionScore_l\}, \ l \in L. \tag{5}$$

#### 3.5.3 TRACEABILITY ANALYSIS

In the context of tracking and understanding the origin and progression of attacks, leveraging the computed dependency paths enables the reverse engineering of a directed graph comprising M-tokens for multi-modal feature fusion. Each node, representing an M-token, consolidates features

amalgamated from diverse modalities and data channels. It is acknowledged that during an attack, alterations in features manifest across various data channels. Consequently, by scrutinizing the variations in different modal data within each M-token, we can identify the data channels crucial to the M-token. This analytical process leads to the establishment of a cross-information domain network attack source tracing pathway.

## 4 EVALUATION OF M-IDAS'S EFFECTIVENESS

In this section, we undertake fifteen intrusion detection tasks to demonstrate the effectiveness of M-IDAS across diverse attack scenarios. Our model is benchmarked against five existing methods, and a thorough comparative analysis is conducted. Furthermore, we delve into interpretative analysis and provide explanations for tracing the origin and progression of attacks.

### 4.1 EXPEIMENT SETUP

#### 4.1.1 DATASET AND DOWNSTREAM TASKS

To evaluate the effectiveness of M-IDAS, we use six categories fifteen attack data and one benign data as the dataset with 187,429 samples, 4.52GB in size and containing 95,963,648 tokens (refer to Appendix A.1), were collected. Unlabeled data was utilized for pre-training, with at most 300 samples randomly selected from each attack class. The dataset was split into training, validation, and testing sets with an 8:1:1 ratio. We conduct evaluation across fifteen intrusion detection tasks shown as follows.

**Task 1-10:** DDoS intrusion detection task aims to identify a spectrum of DDoS attacks prevalent in the IoT environment. Notably, within the domain of IoT security, these attacks constitute the predominant category of threats. Tasks 1-10 delineate various DDoS attack types, including, but not limited to, SYN Flood, HTTP Flood, and other related modalities.
**Task 11:** Password attack detection aims to detect brute force login password attacks, etc.
**Task 12:** XSS attack detection aims to detect malicious XSS scripts in the network.
**Task 13:** MITM attack detection aims to identify and counteract man-in-the-middle attacks, thereby ensuring that information remains both confidential and unaltered during transmission.
**Task 14:** Port scan detection aims to identify indications of port scanning activity, a precursor often utilized to gauge potential points of intrusion prior to executing more sophisticated attacks.
**Task 15:** SQL injection detection aims to the identification of injection attacks targeting service databases. Typically, these attacks are executed to illicitly gain access and exfiltrate sensitive data.

#### 4.1.2 EVALUATION METRICS AND SETTINGS

We utilize four conventional metrics: Accuracy (ACC), Precision (PR), Recall (RC), and the F1 score in the evaluation. To account for potential biases stemming from data category imbalances, we employ the Macro Average methodology, which computes the mean value of ACC, PR, RC, and F1 across each category. Multiple cross-validation is employed to ascertain the evaluation result. Pre-training employed parameters with a batch size of 32, a cumulative step count of 800,000, a learning rate of $2 * 10^{-5}$, and a warmup ratio of 0.1. Subsequent fine-tuning, performed using the AdamW optimizer over 10 epochs, maintained a consistent batch size of 32 and a dropout rate of 0.5. The experiments were conducted using PyTorch 1.12.1 and UER-py (Zhao et al., 2019), executed on NVIDIA A100 GPU hardware.

### 4.2 COMPARISON WITH OTHER METHODS

We compare M-IDAS with various state-of-the-art (SOTA) methods for intrusion detection, including (1) fingerprint construction method: FlowPrint (Van Ede et al., 2020), (2) detection method based on host behavior: CSysCall (El Khairi et al., 2022), (3) detection method based on network traffic: Kitsune (Mirsky et al., 2018), (4) machine learning method based on multi-modal intrusion data: MNAD (Xu et al., 2022), and (5) deep learning method based on multi-modal intrusion data:

Table 1: Comparision Results on Fifteen Intrusion Detection Tasks

| Methods | Task1-10 | | | | Task11 | | | | Task12 | | | |
|---|---|---|---|---|---|---|---|---|---|---|---|---|
| | ACC | RF | PR | F1 | ACC | RF | PR | F1 | ACC | RF | PR | F1 |
| FlowPrint (2020) | 0.605 | 0.891 | 0.637 | 0.743 | 0.835 | 0.901 | 0.786 | 0.839 | 0.751 | 0.757 | 0.624 | 0.684 |
| CSysCall (2022) | 0.760 | 0.747 | 0.598 | 0.664 | 0.715 | 0.673 | 0.598 | 0.633 | 0.718 | 0.807 | 0.610 | 0.695 |
| Kitsune (2018) | 0.752 | 0.835 | 0.767 | 0.800 | 0.766 | 0.843 | 0.843 | 0.843 | 0.849 | 0.810 | 0.842 | 0.826 |
| NMAD (2022) | 0.703 | 0.797 | 0.643 | 0.712 | 0.871 | 0.816 | 0.789 | 0.802 | 0.819 | 0.840 | 0.778 | 0.808 |
| M-CAD (2023) | 0.862 | 0.861 | 0.830 | 0.845 | 0.851 | 0.869 | 0.736 | 0.797 | 0.894 | 0.841 | 0.740 | 0.787 |
| M-IDAS (ours) | **0.990** | **0.970** | **0.991** | **0.980** | **0.989** | **0.988** | **0.975** | **0.981** | **0.992** | **0.988** | **0.984** | **0.986** |
| Methods | Task13 | | | | Task14 | | | | Task15 | | | |
| | ACC | RF | PR | F1 | ACC | RF | PR | F1 | ACC | RF | PR | F1 |
| FlowPrint (2020) | 0.853 | 0.826 | 0.517 | 0.636 | 0.836 | 0.901 | 0.732 | 0.808 | 0.740 | 0.879 | 0.628 | 0.733 |
| CSysCall (2022) | 0.788 | 0.711 | 0.656 | 0.682 | 0.844 | 0.629 | 0.552 | 0.588 | 0.748 | 0.669 | 0.555 | 0.607 |
| Kitsune (2018) | 0.824 | 0.806 | 0.758 | 0.781 | 0.762 | 0.845 | 0.810 | 0.827 | 0.799 | 0.885 | 0.702 | 0.783 |
| NMAD (2022) | 0.708 | 0.833 | 0.685 | 0.752 | 0.868 | 0.757 | 0.817 | 0.786 | 0.906 | 0.780 | 0.810 | 0.795 |
| M-CAD (2023) | 0.857 | 0.824 | 0.794 | 0.809 | 0.896 | 0.901 | 0.808 | 0.852 | 0.886 | 0.806 | 0.828 | 0.817 |
| M-IDAS (ours) | **0.969** | **0.967** | **0.965** | **0.966** | **0.994** | **0.954** | **0.990** | **0.972** | **0.963** | **0.985** | **0.968** | **0.976** |

M-CAD (Bahadoripour et al., 2023). The experimental results are shown in Table 1. Additional comparison studies can be found on our website [4]

**Task1-10:** M-IDAS outperforms all other methods, showcasing a notable performance advantage. Our model achieves a remarkable improvement of 18.0% and 13.5%, respectively, compared to state-of-the-art techniques like Kitsune and NMAD across task1 to task10. Kitsune relies solely on single-modal data (network traffic) to build a multi-dimensional deep learning intrusion detection model, whereas M-IDAS effectively leverages multiple modal data sources, simultaneously learning interrelationships among fused features from four different types of detection data.

**Task11:** M-IDAS outperforms the current state-of-the-art model by 15.2% in Password attack detection. Our approach excels not only in learning traffic features but also demonstrates a robust capability in multi-modal feature learning and fusion.

**Task12:** In Task 12 (XSS attack detection), our method outperforms all other methods. With an F1 score of 98.6%, our performance surpasses Kitsune. These results highlight the superiority of multi-modal data detection over single-modal network traffic detection.

**Task13:** M-IDAS demonstrates a significant 15.7% improvement over the current best result obtained by M-CAD. M-CAD simply use a features splicing network to learn the fusion features, while we exploit the intrinsic relationship of fused multi-modal features to achieve better classification.

**Task14:** M-IDAS achieves the best performance on F1 as 97.2% on Portscan detection by deeply representing multi-modal soure data. The integration of traffic behavior, device status, and system status contributes to enhanced detection effectiveness.

**Task15:** Regarding SQL injection detection (see Table 1), our method outperforms all others. M-IDAS attains the highest F1 score at 97.6%, demonstrating its robustness in identifying diverse intrusion types, even in the presence of injection attacks.

## 4.3 INTERPRETATIVE ANALYSIS

We performed a random selection of 50 samples from each category of attack samples to establish the ground truth dataset of interpretative analysis. Subsequently, we conducted intrusion samples detection and computed the model's attention scores associated with each intrusion sample. These attention scores were then utilized to construct an attention dependency graph. From this graph, we extracted the fused multi-modal features dependency paths with the highest weights, which were considered as the attack dependency path maps. Following the acquisition of the attack dependency path maps, we conducted an in-depth analysis of the fused multi-modal node data associated with each node. By comparing the variations in node data before and after each domian channel of intrusion data, we identified the key data channels within the fused multi-modal nodes. This approach allowed us to achieve cross-domain multimodal attack traceability and explainability analysis. The results of our evaluation are visually presented in Figure 4. Remarkably, our analysis achieved an

---

[4] https://sites.google.com/view/site-midas/

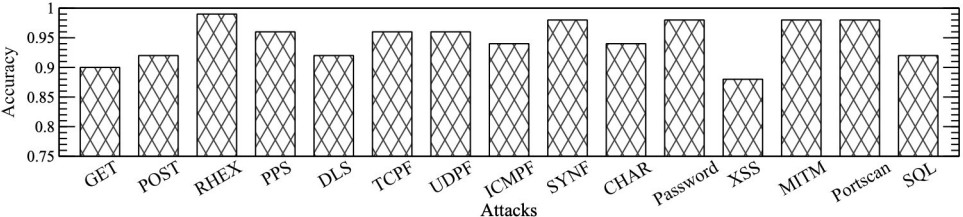

Figure 4: The accuracy of interpretative analysis across fifteen attacks

accuracy rate of 94.4% in providing explanations and traceability analysis for six categories attack containg 15 distinct attack behaviors.

## 5    RELATED WORK

**Multi-Modal Learning.** Multi-modal utilizing information from different modalities can provide more robust and comprehensive insights than relying on a single data source (Gao et al., 2020). For instance, The integration of visual and radar-based sensory systems can enhance safety in high-risk vehicular environments Prakash et al. (2021). The current challenge lies in effectively combining and representing the diverse modalities to achieve superior performance in tasks like classification, regression, or generation (Jiang et al., 2023; Liu et al., 2022). Fusion techniques, for instance, involve extracting features from each modality separately and combining them in the decision-making phase (Li et al., 2022b). Alignment and attention mechanisms have been employed to weigh the importance of various modalities depending on context (Zhu et al., 2023; Vaishnav & Serre, 2022). **Intrusion Detection.** Consequently, numerous researchers have delved into multi-modal intrusion detection methodologies. This includes strategies that leverage multi-modal fingerprints (Spinoulas et al., 2021; Gomez-Barrero et al., 2018) and data from multi-modal sensors (Wagan et al., 2023). These multi-modal approaches cover a broad spectrum of attack scenarios, from general ones (Xu et al., 2023) to those specifically targeting industrial systems (Bahadoripour et al., 2023), image attack detection (Kumari & Singh, 2022) and threats detection (Zhang et al., 2019).

## 6    DISCUSSION AND FUTURE WORK

In this section, we delve into the constraints of this work and explore its possible impact on spurring future research in the domain. The explanatory analysis omits crucial decision-making factors in certain situations. For instance, XSS attacks are predominantly marked by the conveyance of malevolent code data. Yet, due to insufficient deep inspection of the network payload, some network traffic went undetected during our analytical procedure. Model pre-training security: while M-IDAS exhibits commendable generalization and robustness across a spectrum of intrusion detection scenarios, it depends on the pure and clean pre-training data. If an adversary intentionally introduces low-frequency subwords as "malicious" embeddings (Kurita et al., 2020), it becomes feasible to produce a compromised pre-trained model embedded with a "backdoor".

## 7    CONLUSION

In this work, we introduce M-IDAS, a novel multi-modal intrusion detection and analytic system. It learns fused representations of multi-modal data features across four categories and six domains during attack events. M-IDAS pre-trains contextual interrelations of these fused features from large-scale unlabeled data, efficiently detects intrusion events across multiple scenarios through simple fine-tuning on small amounts of task-specific labeled data, and provides explanations and analyses of attack paths and cross-domain key factors for decision-making. Extensive evaluations on six categories of fifteen attack datasets demonstrate M-IDAS's exemplary generalization and robustness, achieving a new state-of-the-art performance with an average accuracy of 98.3% across 15 intrusion detection tasks, including DDoS, Password, Portscan, MITM, XSS, SQLI. Future investigations will focus on M-IDAS's ability to predict new intrusion samples and resist sample attacks.

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
