# A APPENDIX

## A.1 ATTACK TAXOMOMY DATASET DETAILS

We examined 15 prevalent attacks in IoT network contexts, categorized into 6 categories, which represent 90% of usual IoT attack scenarios. We also carried out recurrent data analysis on these. The attacks are made up of 10 different DDOS variants and 5 other standard attack types. Attack dataset details seen in Table 2.

Table 2: Attack Dataset Details

| Attack | | | Description | Sample# | Token# |
|---|---|---|---|---|---|
| DDoS | | GET | GET Flood | 400 | 204,800 |
| | | POST | GET Flood | 400 | 204,800 |
| | | RHEX | Random HEX | 400 | 204,800 |
| | | PPS | Only 'GET / HTTP/1.1\r\n\r\n' | 400 | 204,800 |
| | | DLS | A New Method of Reading data slowly | 400 | 204,800 |
| | | TCPF | TCP Flood Bypass | 400 | 204,800 |
| | | UDPF | UDP Flood Bypass | 400 | 204,800 |
| | | ICMPF | Icmp echo request flood | 400 | 204,800 |
| | | SYNF | SYN Flood | 400 | 204,800 |
| | | CHAR | Chargen Amplification | 400 | 204,800 |
| Password | | | attempting to guess or crack the device's password to gain unauthorized access. | 400 | 204,800 |
| XSS | | | injecting malicious code into a website or web application | 400 | 204,800 |
| Man-in-the-middle | | | intercepts and manipulates communication between two parties secretly | 400 | 204,800 |
| Port scan | | | attempt to discover open ports on a computer or network | 400 | 204,800 |
| SQL Injection | | | Injecting malicious SQL code into input to compromise and manipulate a database. | 400 | 204,800 |
| Benign | | | normal and non-malicious data | 181,429 | 92,891,648 |

During the intrusion detection data feature phase, we handled data attributes spanning 4 modalities over 6 domains. This encompassed 80 dimensions of features derived from network traffic, 35 dimensions of features from system event logs, 23 dimensions of features from performance indicators, and 15 dimensions relating features to IoT device status. The features extracted details seen in Table 3.

Table 3: The Features of Intrusion Data

| Category | Domain | Feature Description | Dimension |
|---|---|---|---|
| Network traffic | | A representation of the data packets moving between devices over a network, reflecting the communication and data transfer patterns. | 80 |
| System Event logs | Proccess infomation | Detailed data about each running application or task on a system, including its state, resource usage, and ownership. | 16 |
| | Execute command log | A record of commands input into the system, capturing user operations and the associated details of each command. | 29 |
| Performance Metrics | CPU Metrics | Quantitative data relating to the processor's performance and usage, such as core utilization, load averages, and context switches. | 11 |
| | Memory Metrics | Indicators of how the system's RAM and cache are being used, encompassing data on total usage, free space, and swap operations. | 12 |
| IoT devices status | | Information on the operational state, connectivity, and performance metrics of interconnected smart devices in a network. | 15 |