# OpenReview forum: "M-IDAS: MULTI-MODAL INTRUSION DETECTION AND ANALYTIC SYSTEM"
_ICLR.cc/2024/Conference — ICLR 2024 Conference Withdrawn Submission_

### Official Review · Reviewer_gzsQ · 2023-10-21

**Soundness:** 1 poor
**Presentation:** 3 good
**Contribution:** 1 poor
**Rating:** 3
**Confidence:** 5

**Summary:**

The paper proposes a novel methodology, reliant on machine learning (ML), to augment the proficiency of intrusion detection systems (IDS). Specifically, the main contribution is "M-IDAS", which is a "novel" IDS which seeks to identify cyberattacks by performing a "multi-modal" analysis by means of various learning algorithms (such as, e.g., transformers). The main intuition is that it is impossible to detect (some) cyberattacks by looking at only one data source (e.g., only the network traffic), and hence M-IDAS seeks to overcome such a limited scope (allegedly adopted by various related works) by proposing to combine the analysis of various data sources (e.g., network traffic AND system logs) into a "single dataset", which is used to train the proposed M-IDAS. Evaluation on a self-collected dataset capturing various information generated by diverse IoT devices shows that M-IDAS outperforms some existing methods for intrusion detection.

## High level comment

I thank the authors for submitting this paper to ICLR24. It was a nice read and I acknowledge that the paper touches a very important and open problem. Indeed, it is daunting, for researchers, to propose methods that combine the information generated by "multi-modal" data sources -- so that the resulting analysis yields actionable insights for the problem of Intrusion Detection in IoT contexts.

However, to my judgment, the paper -- while proposing a solution that "probably works", and whose code and data are going to be publicly disclosed -- does not represent a worthy contribution for ICLR. Indeed, my best observation is that this paper is more suited for an applied ML venue rather than for ICLR, whose goal is mostly to advance our current understanding of learning methods. I have outlined above the major weaknesses that, in my opinion, affect the paper. To align my review with ICLR's guidelines, I summarize the following points:

* Originality: mediocre. While rare, there exists papers that propose "multi-modal" solutions devoted to IDS. However, the specific way this problem is addressed in this paper is "novel".
* Quality: poor. The evaluation methodology and the design of M-IDAS omit important details that prevent one from pinpointing its scientific soundness.
* Clarity: good. The paper is well written and easy to follow, and the schematics/tables are useful.
* Significance: poor. This is a combination of the mediocre originality and poor quality. As it currently is, I doubt that this paper represents a valuable addition to the state-of-the-art.

In what follows, I will list the "Strenghts", and then list and carefully elaborate the many "Weaknesses" that justify my low score.

**Strengths:**

+ The code and data will be released
+ The text is readable
+ The results are promising
+ The comparison entails various methods (some of which being very good already)
+ The proposed solution addresses an open problem (both in research and in practice)

**Weaknesses:**

## Overview

- The overall contribution to the state of the art is "unclear"
- There is a poor treatment of prior works (i.e., the outlined "challenges" are not really challenges)
- The proposed methodology is not sound, and most design choices are arbitrary
- The experiments are done on a self-collected dataset
- The comparison with state of the art methods is unclear
- The evaluation omits important details that are relevant for a significant contribution in this domain

In what follows, I will better elaborate all of the abovementioned weaknesses. I endorse the authors to read my comments carefully -- most of which will be provided with direct quotations of the paper, and/or listing of references that support the observation.


### Unclear contribution

None of the three claimed contributions (at the end of the Introduction) are compelling -- at least from the perspective of research oriented on machine learning based intrusion detection. Let me express my concerns directly:

> We summarize a comprehensive taxonomy for IoT-related intrusion and implement an attack replay toolkit1, collected 15 categories of attacks across six domains at 4.52 GB.

Given that (i) there are already many ```taxonomies for IoT intrusions``` (e.g., [M, N, O]) , and that (ii) there are already ```attack replay toolkits``` (e.g., [P, Q]), I wonder how relevant such a contribution is. Plus, what makes the it a ```comprehensive taxonomy```? On what basis is this adjective used?

> We propose a pre-training approach (M-IDAS) for intrusion detection. It utilizes multiview features, leveraging multi-modal fusion to learn attack feature representation across multiple domains. Employing large-scale unlabeled intrusion data, M-IDAS learns the relationship among the fused features achieving an impressive average accuracy at 98.3% across diverse tasks.

Claiming that 98.3% average accuracy is ```impressive``` is an overstatement. There are literally thousands of papers that achieve similar performance metrics (see, e.g., [L]). Hence, claiming this as a strong point of M-IDAS is not a compelling argument.

> We newly propose cross-domain multi-modal intrusion traceability and explainability analytics, which extracts key fusion features and dependency trajectories from the attention score and offers insights into which domain influences the decision-making process.

Merely ```proposing``` something is not a contribution per-se. The question is: does this proposal actually work well? For instance, how was this proposed method assessed, and what are its results? Without such evidence, it is impossible to gauge the quality of such a contribution.

### Poor "Challenges"

In the Introduction, the paper states three challenges, and the alleged contribution seeks to overcome all of these. However, I disagree with the arguments provided in the paper. Specifically, I am not convinced either by the challenges themselves, nor by the way they are tackled in this paper. Let me elaborate by directly quoting the relevant parts of the text:

> Traditional intrusion detection methods, reliant on a singular data source, encounter significant limitations. They struggle to cross-verify malicious activities, leading to an incomplete understanding of the attack landscape and increased vulnerability to evasion by sophisticated threats. This limitation stems from the inadequacy of individual data sets to comprehensively depict cyber-attack trends (CH1).

I disagree. As written, it appears that this "limitation" is true universally. Having worked in a SOC, in practice, IDS (or rather SIEM or SOAR) do consider multiple data sources for their analyses (see, e.g., [A,B,C,D,J]). The fact that datasets have limitations is mostly a problem "in research", and specifically to those research whose proposals are based on publicly available data (e.g., [E]). However, there are even research papers that overcome this limitation in the specific field of machine learning: an example is, e.g., [F]. Hence, while I do agree that it is challenging for research papers to carry out evaluations that consider multiple perspectives (due to the lack of publicly available data) such a challenge only pertains to a strict subset of research efforts (which can be still overcome)

> While multi-modal learning has shown breakthroughs in various
fields its application in intrusion detection remains limited. Existing work, although applying multi-modal features, often focuses solely on network traffic data, lacking the breadth required for comprehensive attack description and cross-modal interpretation capabilities (CH3).

I disagree. As pointed out in my previously referenced works, there are papers which achieve outstanding results in recognizing attacks even by just looking at the "network perspective". Plus, there are also works that do consider multiple perspectives, and do so efficiently. Finally, w.r.t. ```comprehensive attack description and cross-modal interpretation capabilities```, I specifically point the authors to [G,H,I]

> To enhance data representation robustness and resolve CH1, we executed fifteen categories of IoT attacks in a controlled setting, collecting data across four modalities spanning six domains.

Such an approach does not ```resolve``` CH1 (actually, it may make it worse). Indeed, what the paper describes is merely that by collecting data from various sources it is possible to "tackle" this problem. However, to ```resolve``` the challenge, one should have publicly released a lot of data that could universally help researchers to carry out similar assessments---and by "a lot" I mean that it should encompass various environments (and not just one). Hence, while I do appreciate the willingness of sharing the data (and code), this will hardly ```resolve``` CH1. Finally, it is unclear what an ```IoT attack``` actually is.

> Finally, to tackle CH3, we present a bidirectional encoder classification model with a multi-head attention mechanism for intrusion detection. This model concurrently utilizes an attention score for cross-domain interpretative analysis. The effectiveness is validated through 15 attack detection tasks across six categories, achieving an average F1 of 97.7%. Ground truth construction for each attack type verifies cross-domain interpretability, with an average accuracy of 94.4%.

Such a description is not very compelling. As I stated above, there are works that propose a "multi-modal" approach to detect intrusions/attacks. So, setting aside that CH3 is not truly a challenge (unless the paper provides evidence that _specific prior works_ have shortcomings, i.e., they cannot detect attacks that can actually be detected by the proposed method), the approach followed to tackle CH3 is not convincing: indeed, the data used to perform the analysis was collected "in-house", thereby raising skepticism about any evaluation that is carried out on the basis of such data (since it may include experimental bias [K,L]).

### Poor choice of testbed

> By analyzing IoT-related attack vectors, we prioritized six primary categories that cumulatively represent approximately 90% of all documented incidents (IBM-Security, 2022). Notably, DDoS attacks, particularly those leveraging the Mirai botnet infrastructure, emerged as the most dominant, constituting an estimated 40%-50% of the total IoT attack landscape

I appreciate the thought of focusing on threats that are more common nowadays, but this is not enough for a sound assessment: indeed, given the plethora of works that studied the in-and-outs of the Mirai botnet [R], the point is "are these attacks a problem to detect today?". In other words, it's true that the Mirai botnet is still active, but whether it represents a problem _for systems that are protected by an IDS_ is a different matter. Hence, if existing IDS can already detect the Mirai botnet (which, AFAIK, they can), then focusing just on this type of attack vector may be inappropriate to provide a convincing testbed.

Of course, things would be **a lot better** if the paper showed that the considered attack (i) cannot be detected via existing techniques, but (ii) it can be detected with M-IDAS. Such an approach would dramatically improve the contribution of this work, both from a methodological perspective (with M-IDAS), and a practical one (with the proposed toolkit)!


### Unclear design steps

>In light of the observations across six categories of attacks, we collect data from four modalities spanning six domains: network traffic, host system event logs, host system performance metrics, and IoT device status, for intrusion detection

I have many concerns with this approach. Essentially, what is being described here is a method that is designed on the specific attacks carried out by Mirai. Hence, I am not surprised that the proposed method will work later (under the assumption that it is evaluated in a similar testbed). Nonetheless, I do not see a clear description about why ```six domains``` were chosen. Even after looking at the Appendix, I only saw the mentioning of these domains, but no justification.

> In the present study, the architectural framework encompasses 12 Transformer blocks, each equipped with 12 attention heads within their respective self-attention layers. The dimensionality of each input token, denoted as H, is fixed at 768, while the total count of input tokens stands at 512.

No justification is provided about why any of these thresholds are chosen.

> After gathering multi-modal data, we accumulate information from various sources, including network traffic, system event logs, performance metrics, and the status of IoT devices. These data sources yield distinct feature sets: 80 dimensions from network traffic, 35 dimensions from system event logs, 23 dimensions from performance metrics, and 15 dimensions from IoT device status3

Again, no justification here (not even in the Appendix). This is problematic, especially in light of recent works showing that some features are not appropriate at all for the sake of ML-assisted intrusion detection. See, e.g, [S].

> Employing time T as the fundamental unit of data representation features corresponding to network traffic, process behavior, execution command patterns, and file operations are extracted within each interval T, where T is set to 10 ms.

Why 10ms? Do note that exactly calibrating similar time-based thresholds is practically daunting. See the case study in [B].

### Poor Evaluation details

Section 4 is focused on the empirical evaluation of M-IDAS. Upon reaching this section, I was intrigued to find that there is mentioning of some sort of comparison between M-IDAS and some state-of-the-art techniques. **I strongly invite the authors to mention this sooner in the paper**, since it would have made the contribution more compelling.

Nevertheless, the provided details for such an evaluation are shallow. In fact, the proposed M-IDAS is compared with 5 prior works which proposed to use some ML methods to detect intrusions. However, while the results in Table 1 do show a substantial improvement over all such prior works, it is unclear how the comparison was carried out _in practice_.

Indeed, the text only reports the following:

> We compare M-IDAS with various state-of-the-art (SOTA) methods for intrusion detection, including (1) fingerprint construction method: FlowPrint (Van Ede et al., 2020), (2) detection method based on host behavior: CSysCall (El Khairi et al., 2022), (3) detection method based on network traffic: Kitsune (Mirsky et al., 2018), (4) machine learning method based on multi-modal intrusion data: MNAD (Xu et al., 2022), and (5) deep learning method based on multi-modal intrusion data: M-CAD (Bahadoripour et al., 2023). The experimental results are shown in Table 1. Additional comparison studies can be found on our website 4

First, upon checking the website (in the second-half of October 2023), there was no mentioning of any ```additional comparison studies``` (and, regardless, I think it is unfair to provide technical details usable for peer-review on a public website). Second, there is no additional information provided (anywhere) about how such prior works have been reimplemented. Although some of these are publicly available, I wonder: where these methods taken "off-the-shelf" and used to perform the exact same analysis as M-IDAS? Because, if so, then the comparison is unfair. For instance, take Kitsune: it is an autoencoder which is trained on a completely different testbed than the one used in this paper. In contrast, M-IDAS is "trained" on a testbed that is specific of Mirai. Hence, I am not surprised at all that M-IDAS outperforms kitsune---but such a large gap is not due to M-IDAS, but rather due to the fact that it was trained on data that is more representative of the test data.

I recommend performing an ablation study that clearly allows one to identify "where" the performance enhancements provided by M-IDAS lie in.

Furthermore, other concerns with the evaluation are: 1) the lack of information on the hardware used to perform the experiments, as well as of the runtime of M-IDAS (for both training and testing it); 2) whether the assessment was repeated enough times to derive statistically significant conclusions or not (for instance, is 0.990 accuracy on Task1-10 due to a single "train:test" assessment, or is it the average among many assessments, each with a different train and test set?); indeed, the text only reports that ```the dataset was split with a 8:1:1 ratio```. I strongly invite the authors to embrace the suggestions of [L].

Lastly, I am confused of how the proposed method should provide benefits in terms of interpretability. The analysis discussed in Section 4.3 is shallow and appears to be done subjectively. Given the many ways to tackle such a problem (e.g., [G]), it is disappointing that there is no mentioning of any related work that could be used to support the method used to validate the analysis.

#### Some additional issues:

* Typo on the title of section 4.1
* The work by Jing Xu occurs twice in the references
* The work by Mirsky is not just an arxiv preprint, but an accepted paper to NDSS18.

#### EXTERNAL REFERENCES

[A]: Bryant, Blake D., and Hossein Saiedian. "Improving SIEM alert metadata aggregation with a novel kill-chain based classification model." Computers & Security 94 (2020): 101817.

[B]: Apruzzese, Giovanni, et al. "The role of machine learning in cybersecurity." Digital Threats: Research and Practice 4.1 (2023): 1-38.

[C]: Muhammad, Adabi Raihan, Parman Sukarno, and Aulia Arif Wardana. "Integrated Security Information and Event Management (SIEM) with Intrusion Detection System (IDS) for Live Analysis based on Machine Learning." Procedia Computer Science 217 (2023): 1406-1415.

[D]: Pierazzi, Fabio, et al. "Scalable architecture for online prioritisation of cyber threats." 2017 9th International Conference on Cyber Conflict (CyCon). IEEE, 2017.

[E]: Mirsky, Yisroel, et al. "Kitsune: an ensemble of autoencoders for online network intrusion detection." NDSS (2018)

[F]: Han, Xueyuan, et al. "Unicorn: Runtime provenance-based detector for advanced persistent threats." NDSS (2020).

[G]: Nadeem, Azqa, et al. "Sok: Explainable machine learning for computer security applications." 2023 IEEE 8th European Symposium on Security and Privacy (EuroS&P). IEEE, 2023.

[H]: Yagemann, Carter, et al. "{ARCUS}: symbolic root cause analysis of exploits in production systems." 30th USENIX Security Symposium (USENIX Security 21). 2021.

[I]: Nadeem, Azqa, et al. "Alert-driven attack graph generation using s-pdfa." IEEE Transactions on Dependable and Secure Computing 19.2 (2021): 731-746.

[J]: De Shon, Markus. "Information Security Analysis as Data Fusion." 2019 22th International Conference on Information Fusion (FUSION). IEEE, 2019.

[K]: Arp, Daniel, et al. "Dos and don'ts of machine learning in computer security." 31st USENIX Security Symposium (USENIX Security 22). 2022.

[L]: Apruzzese, Giovanni, Pavel Laskov, and Johannes Schneider. "SoK: Pragmatic Assessment of Machine Learning for Network Intrusion Detection." IEEE EuroS&P (2023)

[M]: Albulayhi, Khalid, et al. "IoT intrusion detection taxonomy, reference architecture, and analyses." Sensors 21.19 (2021): 6432.

[N]: Hindy, Hanan, et al. "A taxonomy of network threats and the effect of current datasets on intrusion detection systems." IEEE Access 8 (2020): 104650-104675.

[O]: Jamalipour, Abbas, and Sarumathi Murali. "A taxonomy of machine-learning-based intrusion detection systems for the internet of things: A survey." IEEE Internet of Things Journal 9.12 (2021): 9444-9466.

[P]: The AVISPA Project Automated Validation of Internet Security Protocols and Applications https://www.avispa-project.org/

[Q]: Neisse, Ricardo, et al. "SecKit: a model-based security toolkit for the internet of things." computers & security 54 (2015): 60-76.

[R]: Antonakakis, Manos, et al. "Understanding the mirai botnet." 26th USENIX security symposium (USENIX Security 17). 2017.

[S]: Liu, Lisa, et al. "Error prevalence in nids datasets: A case study on cic-ids-2017 and cse-cic-ids-2018." 2022 IEEE Conference on Communications and Network Security (CNS). IEEE, 2022.

**Questions:**

I invite the authors to clearly point out any mistakes in my review (should they find any): if provided with compelling evidence, I will revise my score.

Nevertheless, the authors should carefully answer the following questions:

Q1) After reading my review, do the authors believe that the proposed paper represents a valuable contribution for the scope of ICLR? If so, why?

Q2) How was the comparison with the state-of-the-art works carried out?

Q3) Have the experiments been repeated many times, or are the numbers reported in the tables the result of just one train/test?

---

### Official Review · Reviewer_w6hm · 2023-10-27

**Soundness:** 2 fair
**Presentation:** 2 fair
**Contribution:** 2 fair
**Rating:** 3
**Confidence:** 4

**Summary:**

The paper addresses a significant issue in the realm of IoT security with a comprehensive approach. However, it would benefit from more detailed technical explanations, empirical validation insights, and considerations for practical implementation and generalizability. Additionally, addressing the questions raised above could further strengthen the paper's contribution to the field.

**Strengths:**

The introduction of M-IDAS, a multi-modal intrusion detection and analytics system, is a contribution to the related field. It encompasses both feature extraction and fusion, pre-training of intrusion detection models, fine-tuning for specific tasks, and interpretive analysis. This comprehensive approach shows promise in addressing the stated challenges.

**Weaknesses:**

1. While the paper presents an overview of the M-IDAS system, it lacks in-depth detail regarding the methodology. Readers might require more information on the specifics of feature extraction, data fusion, and the architecture of the proposed model. A more in-depth technical explanation is essential for understanding the approach.
2. The paper mentions achieving impressive results, but it does not provide a comprehensive discussion of the empirical validation process. A more detailed presentation of the experimental setup, data sources, and implementations would enhance the transparency of the research.

**Questions:**

1. How feasible is the implementation of the proposed system, M-IDAS, in practical IoT environments? Is it scalable and adaptable to real-world scenarios, or does it primarily serve an academic research purpose?

2. Given the strong focus on IoT security, what considerations have been made regarding data privacy and security within the M-IDAS system? How can it handle and protect sensitive information in an IoT ecosystem?

3. What are the hardware and computational resource requirements for implementing M-IDAS? Will it be accessible to organizations or individuals with limited resources?

4. While the paper mentions the effectiveness of M-IDAS across various attack scenarios, it's essential to discuss its potential generalizability and applicability in diverse IoT contexts. Can this system be applied to different industries, such as healthcare, industrial control, or smart cities?

5. Will the research team make the M-IDAS system, including its tools and data, available to the wider research community or industry for further evaluation and development?

---

### Official Review · Reviewer_C9Rg · 2023-10-30

**Soundness:** 3 good
**Presentation:** 3 good
**Contribution:** 2 fair
**Rating:** 3
**Confidence:** 3

**Summary:**

The paper discusses the limitations of using single-modal data for attack detection and proposes a new method called M-IDAS based on multi-modal data and transformer technology. M-IDAS utilizes modal fusion to integrate data from different sources and pre-trains a model on unlabeled data to represent data behaviors. Then, the model can then be fine-tuned with a few labeled data, achieving high performance in intrusion detection. The paper also provides traceability and interpretative insights into network attack behaviors, offering a profound understanding of the network attack process.

**Strengths:**

- Great topic, for some complex attacks, single source data may not be enough to track complete attack information. Also given the very different formats of the different source data, merging the information would be challenging.

- Evaluated on different attacks.

**Weaknesses:**

- Motivation is unclear and not well discussed. Although the paper claims that single-source data does not contain as much information as multi-source data, which may help detect complicated attacks, it does not give specific examples. For common attacks, single-source data is sufficient. For example, the DDOS attack mentioned in the paper could be detected by network logs alone. Using multi-source data is unnecessary and may even introduce more noise. Therefore, it would be nice if this paper could discuss more about the specific cases where multi-source data performs better than single-source data, give some case studies, and evaluate on those specific cases.

- Lack of technical contribution. After reading the entire paper, if I understand it correctly, the proposed method is based entirely on existing methods. I'm a little concerned because it also doesn't look like new CHALLENGES were encountered and solved during the implementation. Some of the challenges I thought were not discussed. For example, security data formats are usually very different from NLP, and in my previous experience, e.g., log data, would have hundreds or even thousands of times larger vocabularies than general NLP tasks. Therefore, we usually need a specific kind of tokenization and abstraction preprocessing. However, this is not mentioned in this paper. Also, merging different data can be difficult. However, it seems the paper does not propose an effective solution for this but pick features by human, which is not automatic, generalizable, hard to justify, and need more ablation study. Also, please clarify the techniques implemented by  existing papers and add citations properly (e.g., I wonder if Masked M-token Mode and ATTENTION-DEPENDENCE GRAPH CONSTRUCTION are implemented by this paper or existing work).

- Lack of discussion of threat model and adaptive attacks. As an attack paper, it would be better if the paper could discuss the attack settings, threat models, and adaptive attacks (e.g., mix DDOS requests with normal requests).

- Since this paper uses its own dataset (I understand that public data is usually unavailable in security research), I would expect to see more dataset details to tell whether or not the simulated data matches the real attack scenarios. As also mentioned in the paper, the security data is imbalanced data, and the benign data would take the majority. So, I wonder how many benign/malicious were included in the datasets. When evaluating how the labels are generated, is there any guarantee for such labeling? I am also concerned about whether randomly picking at most 300 samples from each attack class could guarantee the integrity of attacks. Also, the paper mentioned benign datasets. When running on such benign datasets, how many false alarms will be produced? Meanwhile, does the testing dataset only contain unknown attacks that are different from the training datasets? If not, could the method generalize to unknown attacks?

- Lack of necessary ablation study and evaluation details. Comparing different embedding models, downstream task models, and existing single-modal and multi-modal methods could be helpful.

**Questions:**

See comments.